# Spatio-temporal dynamics of pathogenic variants associated with monogenic disorders reconstructed with ancient DNA

**Draga Toncheva**[1,2]*, **Maria Marinova**[3], **Plamenka Borovska**[4], **Dimitar Serbezov**[1]

**1** Department of Medical Genetics, Medical Faculty, Medical University-Sofia, Sofia, Bulgaria, **2** Bulgarian Academy of Sciences, Sofia, Bulgaria, **3** Department of Computer systems and Technologies, Faculty of Electronics and Automation, Technical University–Sofia, Branch Plovdiv, Bulgaria, **4** Department of Informatics, Faculty of Applied Mathematics and Informatics, Technical University of Sofia, Sofia, Bulgaria

* dragatoncheva@gmail.com

**Data Availability Statement:** The data underlying the results presented in the study are available from David Reich Lab (https://reich.hms.harvard.edu/datasets).

## Abstract

Genetic disease burden in ancient communities has barely been evaluated despite an ever expanding body of ancient genomes becoming available. In this study, we inspect 2729 publicly available ancient genomes (100 BP—52000 BP) for the presence of pathogenic variants in 32643 disease-associated loci. We base our subsequent analyses on 19 variants in seven genes—*PAH*, *EDAR*, *F11*, *HBB*, *LRRK2*, *SLC12A6* and *MAOA*, associated with monogenic diseases and with well-established pathogenic impact in contemporary populations. We determine 230 homozygote genotypes of these variants in the screened 2729 ancient DNA samples. Eleven of these are in the *PAH* gene (126 ancient samples in total), a gene associated with the condition phenylketonuria in modern populations. The variants examined seem to show varying dynamics over the last 10000 years, some exhibiting a single upsurge in frequency and subsequently disappearing, while others maintain high frequency levels (compared to contemporary population frequencies) over long time periods. The geographic distribution and age of the ancient DNA samples with established pathogenic variants suggests multiple independent origin of these variants. Comparison of estimates of the geographic prevalence of these variants from ancient and contemporary data show discontinuity in their prevalence and supports their recurrent emergence. The oldest samples in which a variant is established might give an indication of their age and place origin, and an *EDAR* gene pathogenic variant was established in a sample estimated to be 33210–32480 calBCE. Knowledge about the historical prevalence of variants causing monogenic disorders provides insight on their emergence, dynamics and spread.

## Introduction

Ancient DNA has in recent years become a valuable tool for addressing key questions about human evolutionary and demographic history. Ancient human DNA can also yield valuable information about the health and disease of ancient communities. The number and distribution of pathogenic alleles causing disease in contemporary populations have however hardly

**Funding:** The authors have received no specific financial support for the research, authorship, and/ or publication of this article.

**Competing interests:** The authors declare no competing interests.

been evaluated in ancient DNA samples. Such effort will indicate the age, place of origin and pattern of spread for many pathogenic variants and thus illuminate their evolutionary past. This will help explain the differences in the prevalence of many common and rare genetic diseases that contemporary human population exhibit.

By analyzing the disease burden of 147 ancient genomes, Berens et al. (2017) estimated that the overall hereditary disease risks of ancient hominins were similar to modern-day humans, but they also observed a temporal trend whereby genomes from the recent past had a higher probability to be healthier than genomes from the deep past [1]. Homozygotes for the P1104A polymorphism of *TYK2* gene are at higher risk to develop clinical forms of tuberculosis, and Kerner et al. (2021) establish that, while the frequency of this pathogenic variant has fluctuated markedly over the last 10,000 years in Europe, it's frequency has decreased dramatically after the Bronze Age [2].

Population sizes and migration rates were small in ancient compared to modern populations. The small effective population sizes would lead to increased levels of homozygosity, including pathogenic alleles in homozygous state, and thus to more efficient selection against these mutations. A large-scale human exome analysis, however, inferring the age of more than one million autosomal single nucleotide variants (SNVs), estimated that approximately 86% of SNVs predicted to be deleterious arose in the past 5000–10000 years [3]. The effect of recent and explosive population growth, combined with weak purifying selection, is the most probable cause of this excess of rare functional variants [4, 5]. It can therefore be speculated that the occurrence of rare diseases has varied substantially among different ancient communities as well as among different time periods.

The majority of rare genetic disease-causing genes however seem to be evolutionarily ancient and ubiquitously expressed in human tissues [6]. The number of genes with variants associated with rare diseases has been considerably increased with the advent of NGS technologies and is currently around 4000 [6]. Rare monogenic diseases (Orphan diseases) are considerable health burden in contemporary populations as the majority of these are marked by severe clinical manifestations, lead to disabilities and often have lethal outcome.

The aim of the present study was to detect and establish the prevalence of pathogenic mutations associated with monogenic diseases in ancient genome-wide DNA samples. This will give an assessment of the disease burden in ancient communities, as well as indicate the temporal dynamics of these pathogenic variants. This will also give an estimate of their minimum age and place of origin. We subsequently compare the prevalence estimates of these mutations in ancient communities to that in contemporary populations.

## Materials and methods

A total of 2729 ancient genome-wide samples were examined [7]. The age distribution of the samples examined (100 BP—15000 BP) is shown on Fig 1.

The DNA sequences were screened for the presence of 369554 disease associated variants taken from the publicly available database of genes and variants associated with human diseases DisGeNet [8]. Genotype data was available for 32644 of these variants in the ancient genome sequences. From these were selected frameshift, missense, splice acceptor, splice donor, splice region, start loss, stop gain and stop loss variants, as all these types of mutations lead to changes in the composition or length of the produced amino acid sequence. We employ the online platform VarSome [9] to select variants for which there is strong evidence of pathogenic impact, i.e. pathogenic computational verdict based on 11 pathogenic predictions from BayesDel-addAF, DANN, DEOGEN2, EIGEN, FATHMM-MK, LIST-S2, M-CAP, MVP, MutationAssessor, MutationTaster and PrimateAI. The exome allele frequencies of these

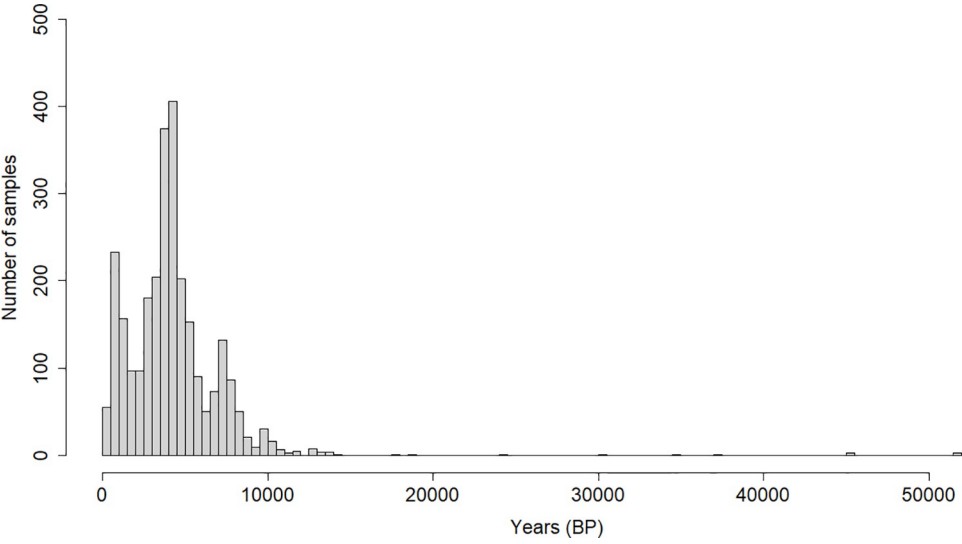

**Fig 1. The age distribution of all samples examined (n = 2729) from 100 BP—52000 BP.**

variants in contemporary populations is taken from Genome Aggregation Database online resource (v.21.1) [10].

## Results

The selection procedure to assemble a set of monogenic disorders associated variants with well-established pathogenic effect yielded 19 unique variants (Table 1). We establish 230

**Table 1. Pathogenic variants associated with monogenic diseases established in homozygote state in the analyzed 2729 ancient genome-wide sequencing samples.**

| dbSNP | Gene | Alleles | oldest | Place of oldest sample containing pathogenic variant | nr. of ancient homozygote genotypes |
|---|---|---|---|---|---|
| rs5030846 | *PAH* | G/A | 9000–7500 BCE | Vasilevka, Ukraine | 18 |
| rs5030850 | *PAH* | G/A | 10420–9450 calBCE | Coquimbo, Los Vilos, Los Rieles, Chile | 11 |
| rs5030851 | *PAH* | G/A | 6773–5886 BCE | Karelia, Yuzhnyy Oleni Ostrov, Russia | 24 |
| rs5030853 | *PAH* | C/A | 1800–1700 BCE | Hazor, Israel | 4 |
| rs5030858 | *PAH* | G/A | 6500–6200 BCE | Northwest Anatolia, Marmara, Turkey | 29 |
| rs5030859 | *PAH* | C/T | 6361–6050 calBCE | Hajduka Vodenica, Serbia | 21 |
| rs76296470 | *PAH* | G/A | 11820–11610 calBCE | Grotte du Bichon, Switzerland | 15 |
| rs5030857 | *PAH* | G/A | 2575–2469 calBCE | Gamo Highlands, Mota Cave, Ethiopia | 1 |
| rs62514952 | *PAH* | C/A | 2500–1900 BCE | Lochenice, Czech Republic | 1 |
| rs62642932 | *PAH* | C/T | 5841–5636 calBCE | Zvejnieki, Latvia | 1 |
| rs62642933 | *PAH* | A/C | 3082–2909 calBCE | Parkhai, Turkmenistan | 1 |
| rs121908456 | *EDAR* | C/A | 1000–800 BCE | Swat Valley, Katelai, Pakistan | 1 |
| rs121908452 | *EDAR* | G/A | 11820–11610 calBCE | Grotte du Bichon, Switzerland | 18 |
| rs121908453 | *EDAR* | C/T | 8210–7836 calBCE | Ganj Dareh, Iran | 20 |
| rs121908450 | *EDAR* | C/T | 33210–32480 calBCE | Goyet cave, Belgium | 45 |
| rs121965063 | *F11* | G/T | 7300–6200 BCE | Motza, Israel | 4 |
| rs63749819 | *HBB* | T/- | 7250–6390 calBCE | Washington State, Kennewick, Columbia River, US | 7 |
| rs121908428 | *SLC12A6* | G/A | 12230–11830 calBCE | Veneto, Villabruna, Italy | 8 |
| rs72554632 | *MAOA* | C/T | 3264–2916 calBCE | Elo Bashi, Russia | 1 |
| Total | | | | | 230 |

homozygote genotypes of these variants in the screened 2729 ancient DNA samples. Eleven of these are in the *PAH* gene (125 ancient samples in total), a gene associated with the condition phenylketonuria in modern populations. Four variants are in the *EDAR* gene associated with hypohidrotic ectodermal dysplasia (83 ancient samples in total), one in the *F11* gene associated factor XI deficiency, a rare bleeding disorder (in 4 samples), one in the *HBB* gene, associated with beta thalassemia, one in the *SLC12A6* gene causing Andermann syndrome (in 8 samples), and one variant in one sample in the *MAOA* gene, associated with mild intellectual disability and behavioral problems. Thirteen ancient genotypes were established to accommodate 2 different pathogenic variants and one 1000 old genotype from Bolivia was even found to accommodate 3 different variants. The oldest sample with detected pathogenic variant is a 33210–32480 calBCE old sample from Goyet cave in Belgium harboring the variant rs121908450 variant in the *EDAR* gene (Table 1). The *PAH* gene rs5030851 variant and *EDAR* gene rs121908452 and rs121908453 variants are established in closely aged samples from vastly different geographical regions, suggesting multiple origin of these variants (Table 1).

The variants examined show varying dynamics over the last 10000 years, some exhibiting a single upsurge in frequency and subsequently disappearing while others maintain high frequency levels (compared to contemporary population frequencies) over long time periods. The temporal dynamics (100 BP-10000 BP) of the established pathogenic variants in the *PAH* gene in ancient samples are shown on Fig 2.

Compared to the overall contemporary population frequencies, as well as the frequencies in different geographical regions [10], the average frequencies of the pathogenic variants in the *PAH* established in ancient samples have a tendency to be higher. Fig 3 shows the overall ancient frequency of the 11 pathogenic variants in the *PAH* gene and the contemporary frequencies from different geographical regions.

## Geographical distribution of pathogenic mutations in ancient communities

The considered pathogenic variants are detected in samples from various geographical regions on different continents. Fig 4 shows the geographical position of the ancient samples in which homozygotes of any of the 11 variants in the *PAH* gene are detected presented, for the sake of clarity, in three time periods (100–3000 BCE, 3000–5000 BCE and 5000–14000 BCE).

## Discussion

The aim of the present study was to examine the publicly available ancient genome-wide data, spanning 100 BP– 52000 BP, for the prevalence of 19 variants that are confirmed to have pathogenic effect in contemporary populations. By doing so, we get an appraisal of the temporal frequency and prevalence dynamics of these variants, and get an indication of their age and place origin.

Eleven of the 19 examined variants are in the phenylketonuria associated *PAH* gene. Some genotypes lead to disease in all environments (high-penetrance Mendelian disorders), disease risk may however only arise by a specific combination between environment and genotype. Phenylketonuria (PKU) manifests only in the presence of mutations that render both copies of the phenylalanine hydroxylase enzyme non-functional and a diet that includes phenylalanine. Low phenylalanine diet may therefore explain how individuals that are homozygous for one or more pathogenic *PAH* gene variant might have been able to survive. In addition, alleles that contribute to disease in modern environments may not have had the same effects in past environments, e.g. fully penetrant mutations in contemporary populations may have had reduced penetrance in ancient human communities. It should also be noted that in general genomic health does not necessarily equate to phenotypic health, especially when comparing populations exhibiting large genetic distances.

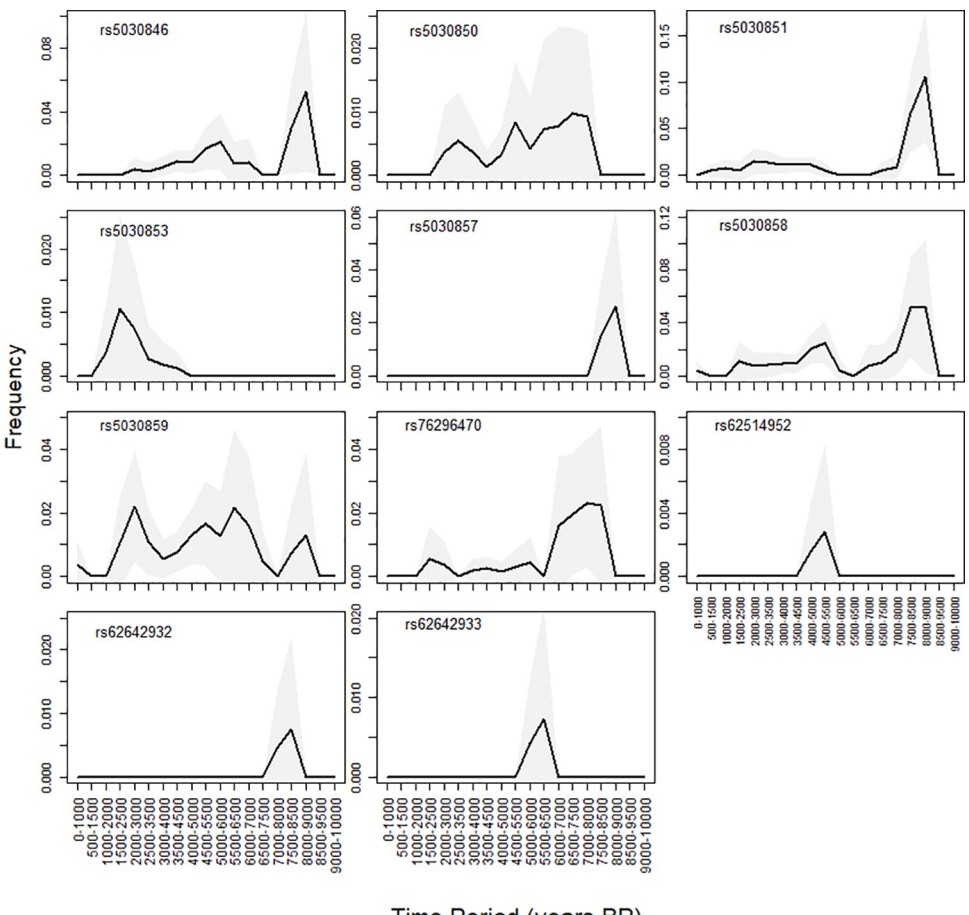

**Fig 2. Temporal dynamics (100 BP- 10000 BP) of the established *PAH* gene pathogenic variants in ancient samples.** The frequency trajectories are plotted using bins of 1,000 years and sliding windows of 500 years. Uncertainty of the frequency estimation is indicated by a gray colored area, representing the normal approximation of the 95% binomial proportion CI.

Information about a variant's age is valuable in interpreting it's functional and selective importance, as has been demonstrated by the relationship between the age and frequency of variants in different geographical regions [12]. The instance furthest back in time a variant has been demonstrated gives an estimate of the minimum age of the variant. We establish the variant rs121908450 of the *EDAR* gene in a DNA sample from Goyet cave in Belgium estimated to be 33210–32480 calBCE (cf. Table 1). Non-recessive deleterious variants have high probability of being purged within a few generations, but both fully and nearly recessive very pathogenic mutations are expected to be purged in small populations [13, 14]. Alleles can also be efficiently removed from populations after bottleneck events [15]. Recent simulation studies have demonstrated that variants annotated as damaging or deleterious were notably absent among older variants (>1,000 generations, or 20000–30000 years) for a given frequency. This also seems consistent with previous findings and theoretical expectations [16, 17]. Parallel to our results for the *EDAR* gene rs121908450 variant, a recent study [2] estimated the age of a *TYK2* gene variant that in homozygous state entails higher risk of developing clinical forms of TB to be ~30,000 years old. Small effective population sizes of ancient human communities facilitates random genetic drift rather than selection, and small populations therefore accumulate

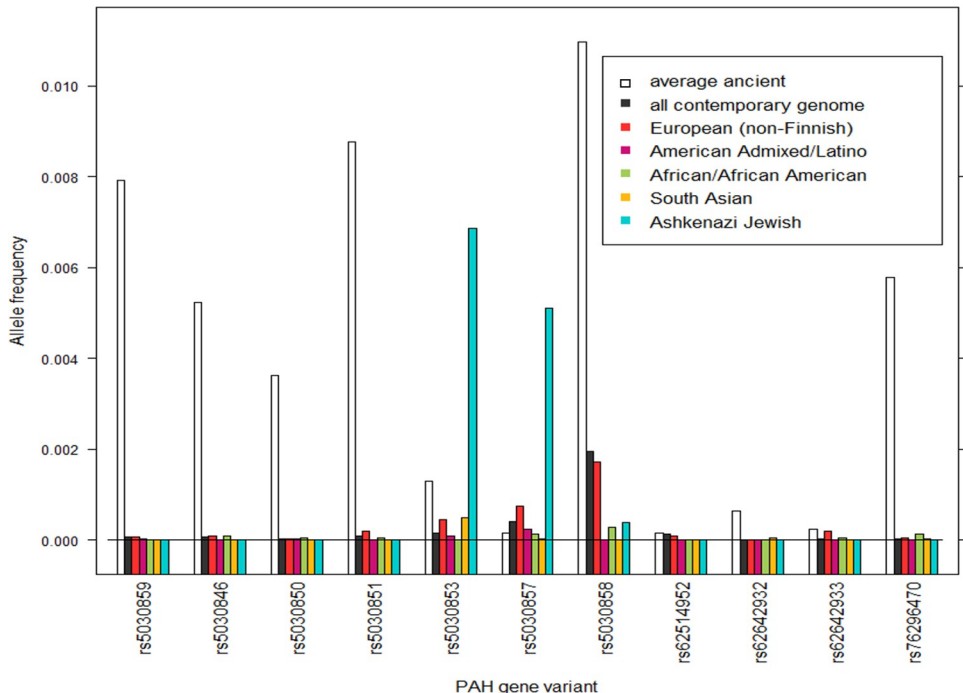

**Fig 3. Average frequencies of the 11 pathogenic *PAH* gene variants estimated from ancient samples (white bars) and contemporary population frequencies (overall and from different geographic regions [10]).**

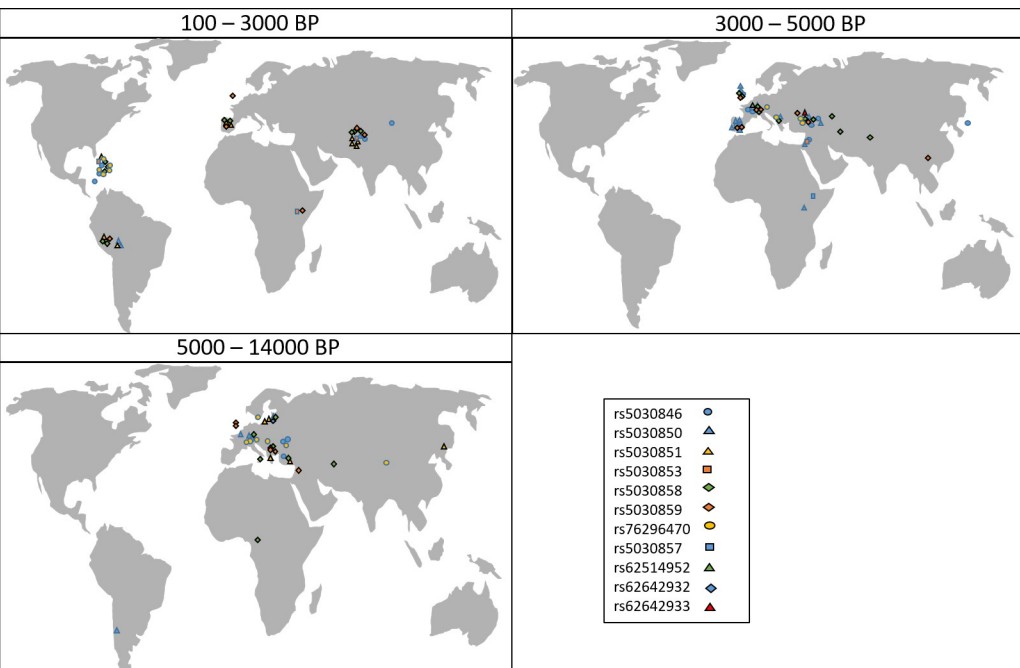

**Fig 4. Geographical position of ancient DNA samples (n = 126) in which homozygote genotypes of the 11 pathogenic variants in the PAH gene were established, allocated to 3 different time periods (BP–Before Present).** Map image obtained from Pixabay [11].

deleterious mutations [18]. Also, the efficiency of purging in small natural populations might be lessened as genetic drift and consanguineous mating generates linkage and identity disequilibrium [19]. This, possibly in combination with a bottleneck event followed by explosive population growth, might have led to high frequency of *PAH* gene rs5030853 and rs5030857 variants estimated in contemporary Ashkenazi Jewish population (cf. Fig 3)

A recent study aimed at dating shared ancestry along with genomic variants in population-scale sequencing data [12] found that, when estimated in independent samples, ages of variants are highly correlated. This agrees with the assumption that, for the majority of alleles, mutations occurred only once in a population's history and thus the age of an allele refers to a fixed point in time [20]. Most individuals carrying an allele would do so through common ancestry if population size is small, as is the case of ancient human communities, and given that the mutation rate is not extremely high. This might be the case for West Eurasian populations where, by around 800 generations ago (16000–24000 years), there is very little structure remaining [12]. We however establish same mutations in ancient DNA genomes from vastly different geographical regions (cf. Fig 4) suggesting that these mutations might have arisen independently multiple times. A comparison between the estimated contemporary and ancient geographical distribution of the considered variants also suggests discontinuity in their prevalence. The *PAH* gene rs5030858 variant, for example, was established in a number of samples from various ancient Asian cites but seems to be absent from contemporary Asians (cf. Figs 3 and 4).

Functional variants are expected to be significantly younger than nonfunctional variants of the same frequency, and variants at high frequency are expected, on average, to be younger than lower-frequency variants [16]. The above considerations might explain why this would not always be the case. We do not finds any relationship between the earliest time a variant was established and it's frequency (Pearson's product-moment correlation r = - 0.251, p = 0.33). Substantial heterogeneity in the relationship between frequency and age have been demonstrated by simulations, suggesting the influence of (often unknown) demographic variables, and this may further be confounded by the mode and strength of selection on particular alleles [12].

The evolutionary approach represents a promising alternative to investigating the genetic sources of present-day disparities between individuals and populations in susceptibility to disease. This approach could however be limited by genetic discontinuity due to temporal and geographic sampling bias, an issue inherent to ancient DNA studies. Inferences about the geographical prevalence of the considered mutations might further be biased by large population replacements [21, 22]. And not least our analyses might have partially been affected by aDNA genome incompleteness and by differential aDNA quality from different geographical regions, again issues inherent to all ancient DNA studies.

## Conclusion

Molecular genetic variation in genome-wide ancient DNA data have been used to evaluate demographic history, divergence and migration patterns of ancient human populations but rarely to assess the genomic health of ancient communities. The prevalence of many common and rare genetic diseases differs among human populations. Diverse genetic and demographic, as well as cultural and environmental, histories give rise to these differences. Synthesizing our growing knowledge of evolutionary history with genetic medicine, while accounting for environmental and social factors, will help to achieve the promise of personalized genomics and permit DNA sequences of individuals to inform clinical decisions. Precision medicine is essentially evolutionary medicine and reaching its full potential necessitates the impact of evolutionary processes also to be considered.

## Supporting information

**S1 Table. Ancient samples analyzed in this study.**
(XLSX)

**S2 Table. Ancient samples containing pathogenic variants considered in this study.**
(XLSX)

## Author Contributions

**Conceptualization:** Draga Toncheva.

**Formal analysis:** Dimitar Serbezov.

**Methodology:** Draga Toncheva, Maria Marinova, Plamenka Borovska, Dimitar Serbezov.

**Project administration:** Draga Toncheva.

**Resources:** Maria Marinova.

**Supervision:** Draga Toncheva.

**Validation:** Draga Toncheva, Dimitar Serbezov.

**Visualization:** Dimitar Serbezov.

**Writing – original draft:** Draga Toncheva, Maria Marinova, Plamenka Borovska, Dimitar Serbezov.

**Writing – review & editing:** Draga Toncheva, Maria Marinova, Plamenka Borovska, Dimitar Serbezov.

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
