## [Decision Letter · Decision Letter 0]

28 Mar 2022

PONE-D-21-35662Spatio-temporal dynamics of pathogenic variants associated with monogenic disorders reconstructed with ancient DNAPLOS ONE

Dear Dr. Draga Toncheva,

Thank you for submitting your manuscript to PLOS ONE. After careful consideration, we feel that it has merit but does not fully meet PLOS ONE’s publication criteria as it currently stands. Therefore, we invite you to submit a revised version of the manuscript that addresses the points raised during the review process.

Please address reviewer comments appropriately. Especially consider the question raised on temporal dynamics. A discussion on sequencing quality in relation to the generated data.  

We look forward to receiving your revised manuscript.

Kind regards,

Klaus Brusgaard

Academic Editor

PLOS ONE

Journal Requirements:

3. We note that Figure 4 in your submission contain [map/satellite] images which may be copyrighted. All PLOS content is published under the Creative Commons Attribution License (CC BY 4.0), which means that the manuscript, images, and Supporting Information files will be freely available online, and any third party is permitted to access, download, copy, distribute, and use these materials in any way, even commercially, with proper attribution. For these reasons, we cannot publish previously copyrighted maps or satellite images created using proprietary data, such as Google software (Google Maps, Street View, and Earth). For more information, see our copyright guidelines: http://journals.plos.org/plosone/s/licenses-and-copyright.

a. You may seek permission from the original copyright holder of Figure 4 to publish the content specifically under the CC BY 4.0 license.  

Reviewers' comments:

Reviewer's Responses to Questions

**Comments to the Author**

1. Is the manuscript technically sound, and do the data support the conclusions?

Reviewer #1: Yes

Reviewer #2: Yes

2. Has the statistical analysis been performed appropriately and rigorously? 

Reviewer #1: Yes

Reviewer #2: Yes

3. Have the authors made all data underlying the findings in their manuscript fully available?

Reviewer #1: Yes

Reviewer #2: No

4. Is the manuscript presented in an intelligible fashion and written in standard English?

Reviewer #1: Yes

Reviewer #2: Yes

5. Review Comments to the Author

Reviewer #1: This is an interesting paper. May be 'at the discussion section ' papers dealing with the human passage between Asia-to Europe/ Africa to Europe. It would be interesting to add at the discussion section 'FVL in Urartian bacck to 3000 BC' as a reference

Reviewer #2: Thanks a lot for this interesting work. I just have my doubts with the proposed temporal dynamics of the traits. Since we still have only access to a limited number of ancient human genomes which are in some periods overrepresented I am wondering how much one can really infer already with the current dataset temporal dynamics. At least I would like to see a statement on the above mentioned limitations. In addition I was missing a supplementary table presenting your findings on single individuum level. I could imagine that some colleagues would be interested to check whether there deposited ancient human remains carried a disease SNP. Thanks in advance for addressing these minor comments.

6. PLOS authors have the option to publish the peer review history of their article (what does this mean?). If published, this will include your full peer review and any attached files.

Reviewer #1: No

Reviewer #2: No

---

## [Author Response · Author response to Decision Letter 0]

4 May 2022

To the Editors of “PLOS ONE”

Dear Sirs,

Thank you for reviewing our manuscript and for letting us resubmit a revised version. We find all comments by the reviewers to be very helpful and constructive. We have taken up their suggestions, and the manuscript has been revised accordingly. Notably, we now clearly acknowledge that sample bias, inherent to all ancient DNA studies, might have affected our inferences on the temporal dynamics and geographic prevalence of the considered mutations. We have now also included supplementary tables that will allow interested researchers to recreate our analyses on the deposited ancient human genome-wide data. We believe that these changes have resulted in a much better version of the manuscript which we hope is more befitting for publication in PLOS ONE.

Sincerely,

Prof. Draga Toncheva

Еditor`s comments:

3. We note that Figure 4 in your submission contain [map/satellite] images which may be copyrighted. All PLOS content is published under the Creative Commons Attribution License (CC BY 4.0), which means that the manuscript, images, and Supporting Information files will be freely available online, and any third party is permitted to access, download, copy, distribute, and use these materials in any way, even commercially, with proper attribution. For these reasons, we cannot publish previously copyrighted maps or satellite images created using proprietary data, such as Google software (Google Maps, Street View, and Earth). For more information, see our copyright guidelines: http://journals.plos.org/plosone/s/licenses-and-copyright.

Our reply: Figure 4 uses a very rudimentary world map that is not copyrighted.

Reviewer #1: This is an interesting paper. May be 'at the discussion section ' papers dealing with the human passage between Asia-to Europe/Africa to Europe.

Our reply: We have now added to references dealing with ancient migrations from Asia and from Africa to Europe that might have impacted our inferences in the Discussion part of the manuscript, Line 206

It would be interesting to add at the discussion section 'FVL in Urartian back to 3000 BC' as a reference

Our reply: We have now referred to this study in the Discussion part of the manuscript, Line 184

Reviewer #2: Thanks a lot for this interesting work. I just have my doubts with the proposed temporal dynamics of the traits. Since we still have only access to a limited number of ancient human genomes which are in some periods overrepresented I am wondering how much one can really infer already with the current dataset temporal dynamics. At least I would like to see a statement on the above mentioned limitations.

Our reply: Temporal as well as geographic discontinuity is inherent to all ancient DNA studies, and we recognize our inferences might be to a certain extend biased. Nevertheless, the considered variants show very differential dynamics indicating that these are at least to a degree independent of the temporal sampling bias. Still, we have modified the text to convey that uncertainty in the inferences about temporal dynamics (Line 31, Line 81, and Line 145). We state in the Discussion part that sample bias might have affected our inferences about the temporal dynamics and geographical prevalence. (Line 206-207)

In addition I was missing a supplementary table presenting your findings on single individuum level. I could imagine that some colleagues would be interested to check whether there deposited ancient human remains carried a disease SNP. Thanks in advance for addressing these minor comments.

Our reply: We have now included as supplementary tables all the ancient samples considered (n=2729), their sex, estimated age, region they were sampled and the published articles they have been used in, Supplementary table S1. Also is included a table with all samples in which any of the considered pathogenic variants were established (n=230), Supplementary table S2.

---

## [Decision Letter · Decision Letter 1]

25 May 2022

Spatio-temporal dynamics of pathogenic variants associated with monogenic disorders reconstructed with ancient DNA

PONE-D-21-35662R1

Dear Dr. Draga Toncheva,

We’re pleased to inform you that your manuscript has been judged scientifically suitable for publication and will be formally accepted for publication once it meets all outstanding technical requirements.

Kind regards,

Klaus Brusgaard

Academic Editor

PLOS ONE

Additional Editor Comments (optional):

Reviewers' comments:

Reviewer's Responses to Questions

**Comments to the Author**

1. If the authors have adequately addressed your comments raised in a previous round of review and you feel that this manuscript is now acceptable for publication, you may indicate that here to bypass the “Comments to the Author” section, enter your conflict of interest statement in the “Confidential to Editor” section, and submit your "Accept" recommendation.

Reviewer #2: All comments have been addressed

2. Is the manuscript technically sound, and do the data support the conclusions?

Reviewer #2: Yes

3. Has the statistical analysis been performed appropriately and rigorously? 

Reviewer #2: I Don't Know

4. Have the authors made all data underlying the findings in their manuscript fully available?

Reviewer #2: Yes

5. Is the manuscript presented in an intelligible fashion and written in standard English?

Reviewer #2: Yes

6. Review Comments to the Author

Reviewer #2: (No Response)

7. PLOS authors have the option to publish the peer review history of their article (what does this mean?). If published, this will include your full peer review and any attached files.

Reviewer #2: No

---

## [Editor Report · Acceptance letter]

14 Jun 2022

PONE-D-21-35662R1 

Spatio-temporal dynamics of pathogenic variants associated with monogenic disorders reconstructed with ancient DNA 

Dear Dr. Toncheva:

I'm pleased to inform you that your manuscript has been deemed suitable for publication in PLOS ONE. Congratulations! Your manuscript is now with our production department. 

Kind regards, 

on behalf of

Dr. Klaus Brusgaard 

Academic Editor

PLOS ONE